# Perturbations in gut microbiota composition in schizophrenia

YiMeng Wang[1☺], SiGuo Bi[2☺], XiaoLong Li[2], YuTao Zhong[2], DongDong Qi[1,2]*

**1** School of Public Health, Inner Mongolia Medical University, Hohhot, Inner Mongolia, China, **2** Clinical Lab, Hulunbuir Third People's Hospital (Hulunbuir Mental Health Center), Yakeshi, Inner Mongolia, China

☺ These authors contributed equally to this work.
\* m13314811012@163.com

**Data Availability Statement:** All relevant data are within the manuscript and its Supporting information files. All raw sequences were deposited in NCBI Sequence Read Archive under accession number PRJNA 1077638 for metagenomics

## Abstract

Schizophrenia is a severe, complex and long-term psychiatric disorder with unclear etiology. Gut microbes influence the central nervous system via the gut-brain axis. Consequently, investigations of the relationship between gut microbes and schizophrenia are warranted. This study involved 29 patients with schizophrenia and 30 age-matched normal controls. After 16S rRNA gene sequencing and whole-genome shotgun metagenomic sequencing, we analyzed microbial diversity, composition, and function. According to 16S rRNA and metagenomic gene sequencing results, patients with schizophrenia had higher abundances of *Clostridium* and *Megasphaera*. Functional analysis showed that sphingolipid, phosphonates and phosphinates, as well as glutamine metabolism were associated with the occurrence and development of schizophrenia. Our data suggest that the gut microbiota exerts an effect on patients with schizophrenia, providing valuable insights into the potential regulation of in the context of this disorder.

## Introduction

Schizophrenia, a chronic psychiatric condition characterized by disturbances in affect, cognition, and behavior, affects approximately 1–2% of the population [1, 2]. The prevalence of schizophrenia varies geographically, and it imposes a significant burden on health worldwide [3]. Schizophrenia is a chronic ailment without a cure. However, patients can manage symptoms, enhance their quality of life, and achieve social adaptation by participating in a rational and comprehensive rehabilitation program [4]. The development of pharmacological interventions for schizophrenia has been slow. Recently developed antipsychotic medications for schizophrenia have comparatively fewer side effects and maintain symptom control [5–7]. However, most patients exhibit in adequate treatment responses. Thus, there is a need to identify novel therapeutic targets. Gut microbes have been implicated in the etiology and progression of various diseases. These microorganisms influence the human immune system and metabolic pathways, thereby modulating conditions such as autism and Alzheimer disease [8, 9]. In 1999, the concept of the "gut-brain axis" was introduced, which describes the intricate interplay between the intestinal tract and the central nervous system [10]. Consequently, gut

sequencing and PRJNA 1077648 for 16S rRNA gene sequencing.

**Funding:** This study was supported by grants from the Inner Mongolia Autonomous Region Science and Technology Innovation Guide project (No. 2021CXYD001) and Inner Mongolia Autonomous Region Science and Technology plan project (No. 2021GG0298). The funders had no role in study design, data collection and analysis, decision to publish, or preparation of the manuscript.

**Competing interests:** The authors have declared that no competing interests exist.

microbes are important regulators in the brain-gut axis. Indeed, the gut microbiota generates and metabolizes diverse compounds, such as neuroactive substances and metabolites, that can alter brain function [11]. The gut microbiota is associated with the onset and progression of schizophrenia; fecal *Clostridium coccoides* exacerbates the onset of schizophrenia [12, 13]. Bacterial infection elicits an inflammatory response in the intestines, leading to the release of inflammatory mediators that can affect the brain via the bloodstream, potentially contributing to the onset and progression of certain mental disorders [14]. Nevertheless, the effects of microbiota disruption on the central nervous system are unknown.

In this study, we hypothesized that the microbiota could serve as a prognostic indicator for patients with schizophrenia. We obtained fecal samples from 29 patients who had been admitted to our hospital. We evaluated microbiota diversity and relative enrichment by 16S rRNA gene sequencing. We then conducted metagenomic analysis to characterize the composition of the bacterial microbiota at the species level and predict their functional pathways in relation to schizophrenia. Our objective was to evaluate the composition and pathways in patients with schizophrenia.

## Materials and methods

### Ethics statement

The study protocol for this study received approval from the ethics committee of Hulunbuir Mental Health Center (No.2021NO.02), and written informed consent was obtained from the participants. To ensure confidentiality, all participant information was anonymized.

### Participants

The 59 participants (30 normal controls and 29 patients with schizophrenia), were recruited between **January and March 2023**. The patients with schizophrenia were attending the Hulunbuir Mental Health Center and had been diagnosed by a clinician using the systematic structured clinical interview method and detailed background information approach detailed in the Diagnostic and Statistical Manual of Mental Disorders, Fifth Edition [15]. The patients showed psychiatric symptoms for $\geq$ 2weeks. The rate of change, as assessed by the Positive and Negative Syndrome Scale (PANSS), was limited to 20% within a 2-week period, and the total PANSS score was required to be $\geq$ 30. Patients with schizophrenia received antipsychotic medications, specifically clozapine and olanzapine, and had been hospitalized for 1 year. The exclusion criteria were: (1) the use of prescription drugs other than antipsychotics during the study period; (2) severe physical illness, pregnancy, or electroconvulsive therapy; (3) contraindications for magnetic resonance imaging or the presence of brain structural abnormalities; (4) history of alcohol/ drug abuse or dependence, or diagnosis of a substance-related mental disorder; (5) history of coma, or neurological disorders; and (6) history of chronic gastrointestinal illness.

The control group consisted of 30 asymptomatic age-matched normal individuals participating in an annual hospital staff examination. The inclusion criteria were: (1) no antibiotic use in the preceding 3 months and absence of diarrhea; (2) absence of chronic illness that could affect gut microbiota stability; (3) body mass index (BMI) of 18–30 kg/m$^2$, indicating a normal metabolic state; (4) no significant gastrointestinal surgery in the past 5 years; (5) no history of head surgery or mental disorder.

## Collection of fecal samples

Fecal samples were obtained with assistance from medical personnel in accordance with ethical guidelines established by the Hulunbuir Mental Health Center. Participants were provided with sterile fecal collectors in advance, and the middle portion of the feces was collected. Fecal samples were promptly stored in a –80˚C freezer until DNA extraction.

## Genomic DNA extraction and amplification

Fecal DNA Kits (catalog no. D4015-02) (Omega Bio-Tek, Norcross, GA, USA) were used to extract total genomic DNA, in accordance with the manufacturer's instructions. DNA samples were stored at –20˚C for further analysis. DNA was quantified using a NanoDrop NC2000 spectrophotometer (Thermo Fisher Scientific, Waltham, MA, USA), and agarose gel electrophoresis was performed to assess DNA quality.

PCR amplification of the v3-v4 region of the bacterial 16s rRNA gene was conducted using the forward primer 338F (5′-ACTCCTACGGGAGGCAGCA-3′) and the reverse primer 806R (5′-ACTCCTACGGGAGGCAGCA-3′). Sample-specific 7-bp barcodes were integrated into the primers to enable multiplex sequencing. PCR mixtures comprised 5 μL of buffer (5×), 0.25 μL of Fast pfu DNA Polymerase (5U/μL), 2 μL of dNTPs (2.5 mM), 1 μL of each forward and reverse primer (10 μM), 1 μL of DNA template, and 14.75 μL of ddH2O. The thermal cycling protocol involved an initial denaturation step at 98˚C for 5 min; 25 cycles of denaturation at 98˚C for 30 s, annealing at 53˚C for 30 s, and extension at 72˚C for 45 s; and a final extension at 72˚C for 5 min. PCR amplicons were purified using Vazyme VAHTSTM DNA Clean Beads (Vazyme, Nanjing, China), and their concentrations were determined using the Quant-iT PicoGreen dsDNA Assay Kit (Invitrogen, Carlsbad, CA, USA).

## Library preparation and 16S rRNA gene sequencing

Amplicons were combined in equal proportions and subjected to paired-end 2 ×250 bp sequencing on the Illumina Miseq platform with the Miseq Reagent Kit v.3 at Shanghai Personal Biotechnology Co., Ltd. (Shanghai, China). Microbiota bioinformatics analysis was conducted using QIIME2 (v. 2023.2.0) in accordance with the manufacturer's instructions with minor adjustments (https://docs.qiime2.org/2023.2/tutorials/). Raw sequence data were processed by merging, quality filtering, barcode cutting, dereplicating, denoising, and chimera removal using the vsearch plugin (v.2.22.1, https://github.com/torognes/vsearch). The resulting sequences were subjected to operational taxonomic unit (OTU) clustering and species taxonomy analysis at a 97% similarity threshold using vsearch. The taxonomic composition of each sample was determined at the kingdom, phylum, class, order, family, genus, and species levels. The sequence with the highest OTU abundance was selected as the representative sequence for a sample. OTU clustering enables diversity indices to be used for OUT analysis, and the sequencing depth can be assessed. Statistical analysis of community structure can be conducted at various taxonomic levels based on taxonomic information. The taxonomic classification of each OTU representative sequence was conducted by searching against the 16S rRNA gene database RDP v.18 (https://mothur.org/wiki/rdp_reference_files), with a confidence threshold of 0.7.

As indices of alpha diversity, we calculated the observed species, Faith's phylogenetic diversity (Faith-PD), Chao1 richness estimator, Pielou's evenness, Shannon diversity index, and Simpson diversity index using usearch v.10.0.240 (https://www.drive5.com/usearch/download.html). The statistical significance of alpha diversity variance across groups was assessed by the Wilcoxon rank-sum test. The Goods_coverage metric was used to indicate sequencing depth. Beta diversity analysis was performed by principal coordinates analysis

(PCoA) to assess differences in species diversity between pairs of samples. Linear discriminant effect size (LEfSe) analysis was conducted online(http://galaxy.biobakery.org/). P-values were determined by the nonparametric factorial Kruskal-Wallis sum-rank test, with the threshold for the logarithmic linear discriminant analysis (LDA) score of discriminative features set to 2.0. The results table underwent additional editing and was subjected to LEfSe visualization to generate histograms and cladograms. Subclasses were compared by pairwise t- test with a significance threshold of 0.05. Clusters of Orthologous Groups of proteins (COG), Kyoto Encyclopedia of Genes and Genomes (KEGG), Metabolic pathways from all domains of life (MetaCyc) pathways, and enzyme category analyses were carried out using PICRUSt v. 2.0 (https://picrust.github.io/picrust/). The significance of pairwise comparisons between groups and the 95% confidence intervals for differences in mean proportions were calculated using STAMP v. 2.1.3 software (https://beikolab.cs.dal.ca/software/STAMP).

### Metagenomic whole-genome shotgun sequencing (WGS)

Whole-genome shotgun sequencing (WGS) enables quantification of gut microbiota at the species level, as well as gene and functional profiling. We selected 12 (6 patients with schizophrenia and 6 normal controls) of the 59 participants whose fecal samples provided sufficient material for metagenomic analysis. From each sample, individual DNA libraries were constructed and sequenced on an MGI-SEQ 2000 (HuaDa Gene, China), resulting in an average of 6 GB of 150 bp ×2 paired-end reads per sample. To ensure data quality, raw reads were processed by cutadapt (v. 1.16) for quality control. After the removal of low-quality segments and reads < 20 bp, BWA (v. 0.7.17-r1188) was utilized to align filtered reads to the human genome (hg38) to deplete host DNA. The remaining the high-quality reads were used for taxonomic profiling with the default parameters.

### Functional profiling of 16S rRNA and metagenomic gene sequencing

Filtered reads were assembled de novo using MEGAHIT (v.1.2.9), and K-mers were iterated from small to large to achieve fewer gaps and long contigs. Functional analyses were performed with BLAST (v. 2.9.0) against the KEGG database, the evolutionary genealogy of genes: Non-supervised Orthologous Groups database (eggNOG), the Carbohydrate-Active enZYmes database (CAZy), the Antibiotic Resistance Genes Database (ARDB) and the Virulence Factor Database (VFDB) using Diamond Pipeline (v. 2.1.6.160).

## Results

### Clinical information

Demographic and sample statistics are listed in Table 1. 29 patients with schizophrenia and 30 normal controls underwent 16S rRNA gene sequencing to investigate species composition, evolutionary relationships, and community diversity in the gut microbiota. Shotgun metagenome sequencing was conducted on 6 patients with schizophrenia and 6 normal controls. We conducted a metagenome-wide association study of the taxonomic and functional composition of the metagenome in the gut microbiota of patients with schizophrenia. A flowchart of the study is presented in Fig 1.

### Sequences information and taxonomic annotation

In total, 2,807,456 valid reads were obtained after 16S rRNA gene sequencing, with an average of 47,584 reads per sample. Clustering of qualified sequences at 97% identity resulted in 15 phyla, 29 classes, 51 orders, 89 families, 175 genera and 170 species. After metagenomics

**Table 1. Demographic and clinical characteristics of the participants.**

| Method | 16S rRNA | | Metagenome | |
|---|---|---|---|---|
| | Schizophrenia | Normal Controls | Schizophrenia | Normal Controls |
| Number(n) | 29 | 30 | 6 | 6 |
| Sex (female/male) | 14/15 | 15/15 | 2/4 | 3/3 |
| Age (years) | 46.28±9.49 | 30.87±5.41 | 42.83±10.83 | 30.50±3.94 |
| BMI | 25.20±4.22 | 24.35±3.26 | 23.63±4.88 | 23.66±3.17 |
| Nation(Han/Mongolia) | 27/2 | 21/9 | 6/0 | 3/3 |
| Subdivision disorder (paranoid/undifferentiated/unknown subtype) | 16/11/2 | - | 3/3/0 | - |

sequencing, 1,344,100,638 clean reads were generated from 1,362,435,978 raw reads; the mean number of reads per sample was 112,008,386. After taxonomic annotation, 134 phyla, 138 classes, 262 orders, 555 families, 2085 genera and 10,324 species were detected.

## Analysis of alpha and beta diversity analysis based on 16S rRNA gene sequencing

To assess alpha diversity, statistical analyses were performed focusing on the alpha diversity index, rank-abundance curve, and species accumulation curve. Alpha diversity reflects the abundance and diversity of microbial communities. The findings indicated a significant increase in Faith-pd among patients with schizophrenia relative to normal controls (S1 Fig). There were few fluctuations in species abundance and evenness among patients with

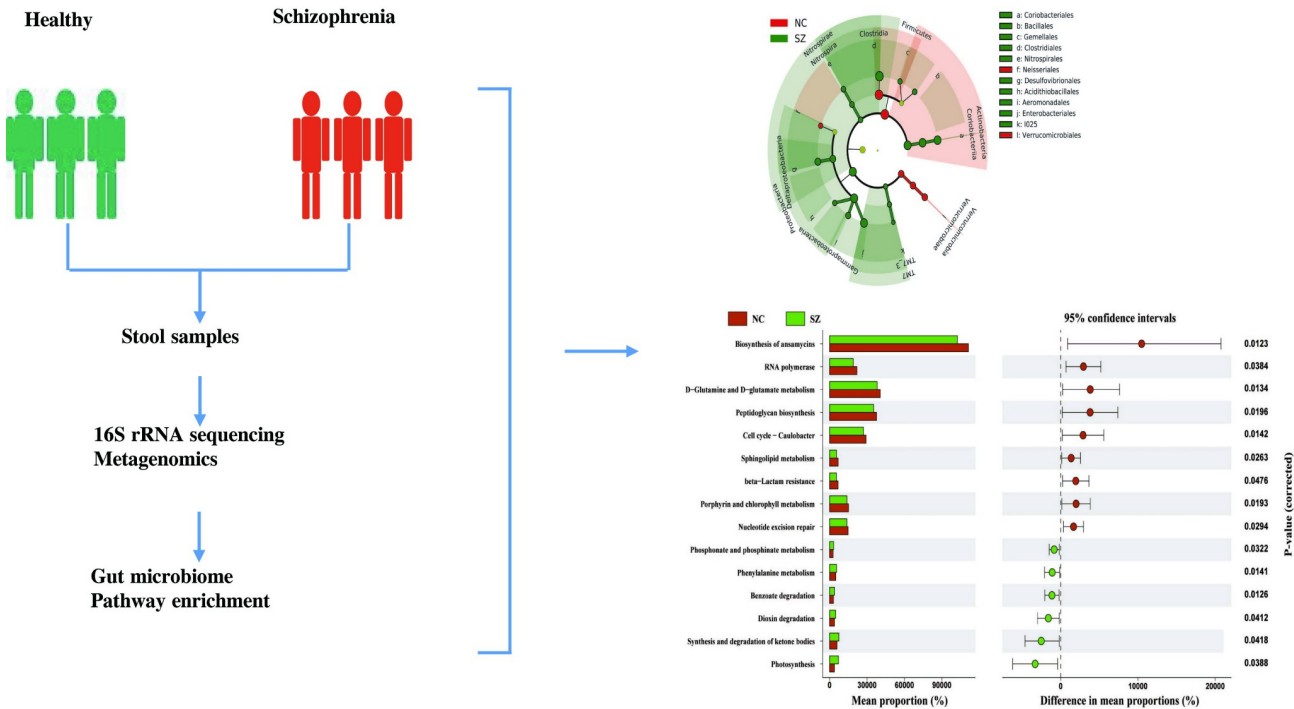

**Fig 1. Flowchart of the study design.** Twenty-nine patients with schizophrenia and 30 normal controls were enrolled. Fecal samples were subjected to 16S rRNA gene sequencing. Clean reads were blasted against the NR database. 6 patients with schizophrenia and 6 normal controls were involved in the metagenomics analysis. Note: SZ: Schizophrenia. NC: Normal Control.

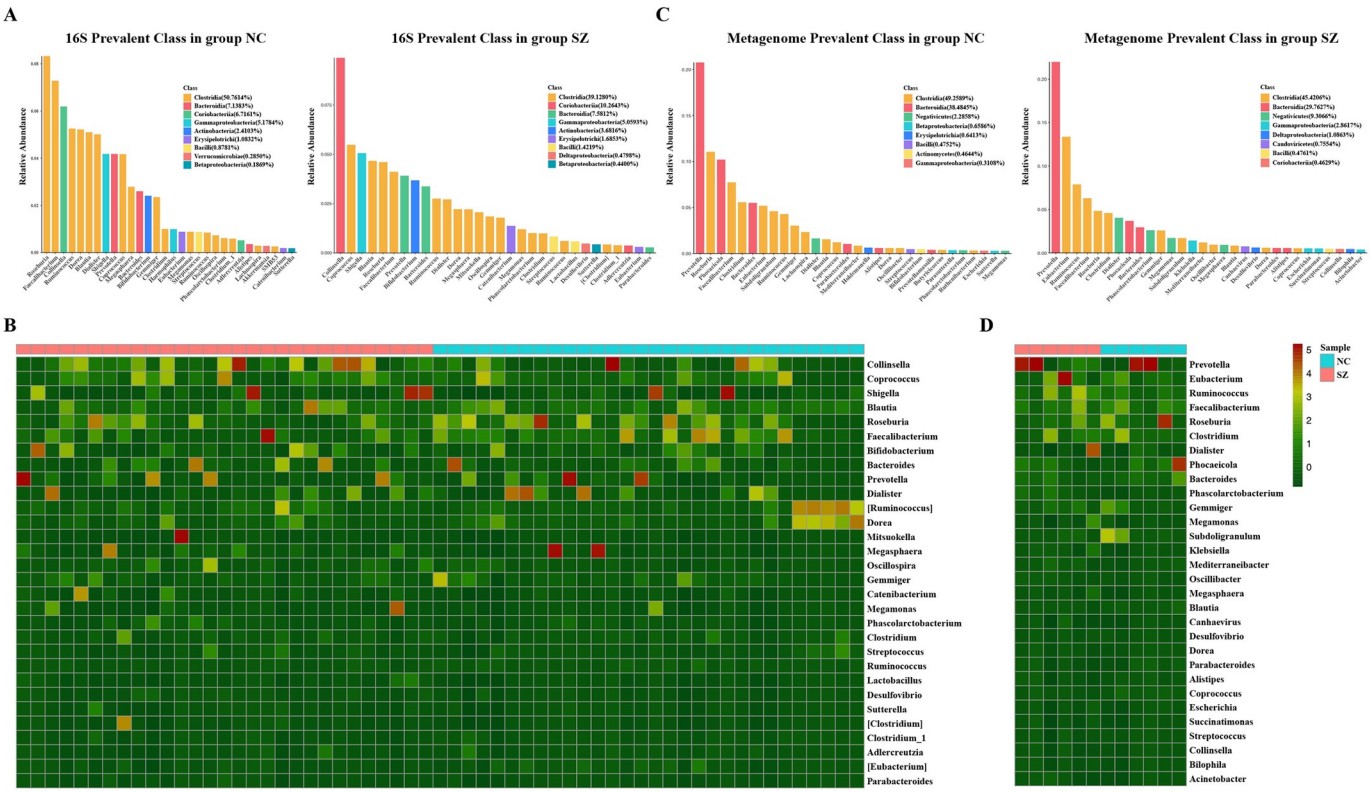

**Fig 2. Taxonomic composition of the fecal microbiota. (A)** Predominant taxa as determined by 16S rRNA gene sequencing. **(B)** Heatmap analysis based on 16S rRNA gene sequencing. Each column represents a sample and each row represents a genus. Different colors represent different relative abundances. **(C)** Predominant taxa based on metagenomic sequencing. **(D)** Heatmap analysis based on metagenomic sequencing. Each column represents a sample and each row represents a genus. Different colors represent different relative abundances. Note: SZ: Schizophrenia. NC: Normal Control.

schizophrenia (S2A Fig). Moreover, the species accumulation curve showed that species richness did not increase in proportion to increasing sample size (S2B Fig), suggesting that 59 were adequate for analysis. Beta diversity was assessed by weighted and unweighted UniFrac PCoA. Microbiota composition differed between patients with schizophrenia and normal controls (S2C and S2D Fig). There was no sex difference in terms of species diversity (S3 Fig).

## Microbiota composition as determined by 16S rRNA gene sequencing

The predominant classes and genera are shown in Fig 2A. The 10 most abundant classes were *Clostridia* (39.2% in group SZ and 50.8% in group NC, note SZ: schizophrenia group, NC: normal control group), *Bacteroidia* (7.6% in group SZ and 7.1% in group NC), *Coriobacteriia* (10.3% in group SZ and 6.7% in group NC), *Gammaproteobacteria* (5.1% in group SZ and 5.2% in group NC), *Actinobacteria* (3.7% in group SZ and 2.4% in group NC), *Erysipelotrichia* (1.7% in group SZ and 1.1% in group NC), *Bacilli* (1.4% in group SZ and 0.8% in group NC), *Betaproteobacteria* (0.4% in group SZ and 0.2 in group NC), *Deltaproteobacteria* (0.5% in group SZ) and *Verrucomicrobiae* (0.3% in group NC). Fig 2B displays a heatmap illustrating the relative abundances of the predominant genera in each sample, as determined by16S rRNA gene sequencing. The microbiota composition differed between group schizophrenia and normal controls, consistent with the results for beta diversity.

## Specific bacteria taxa revealed by the linear discriminant analysis (LDA) effect size (LEfSe) analysis

To identify biologically biological important taxa, we performed LEfSe analysis with a logarithmic LDA score cutoff of $\geq 2.0$ and a P-value $\leq 0.05$. Among the predominant genera, *Mogibacterium* and *Clostridium* showed higher relative abundances in schizophrenia patients, whereas *Lactococcus* and *Akkermansia* were more abundant in normal controls (Kruskal-Wallis, P<0.05) (Fig 3A). As indicated by the cladograms in Fig 3B, *Coriobacteriales*, *Bacillales*, and *Gemellales* were significantly enriched in schizophrenia patients, whereas *Neisseriales* and *Verrucomicrobiales* were significantly more abundant in normal controls.

## Microbiota composition as determined by metagenomic sequencing

Gut microbiota samples from 12 participants (6 normal controls and 6 patients with schizophrenia) were subjected to metagenomic shotgun sequencing. The predominant phyla and genera are shown in Fig 2C. The 12 most abundant phyla were *Clostridia* (45.4% in group SZ and 49.3% in group NC), *Bacteroidia* (29.7% in group SZ and 38.5% in group NC), *Negativiautes* (9.3% in group SZ and 2.2% in group NC), *Bacilli* (0.48% in group SZ and 0.47% in group NC), *Gammaproteobacteria* (2.7% in group SZ and 0.3% in group NC), *Betaproteobacteria* (0.7% in group NC), *Erysipelotrichia* (0.6% in group NC), *Actinomycetes* (0.5% in group NC), *Gammaproteobacteria* (2.8% in group SZ), *Deltaproteobacteria* (1.1% in group SZ), *Caudoviricetes* (0.8% in group SZ), and *Coriobacteriia* (0.5% in group SZ). Fig 2D represents a heatmap showing the relative abundances of the predominant genera in each sample. As shown in Fig 3C, among the predominant genera, *Ruminococcus*, *Phascolarctobacterium*, *Megasphaera*, and *Clostridium* exhibited higher relative abundances in schizophrenia patients, whereas *Lachnospira*, *Coprococcus*, and *Bacteroides* were more abundant in normal controls (Wilcoxon rank-sun test, P< 0.05). The taxa that explained most of the differences were identified by LEfSe. *Clostridium*, *Megamonas*, and *Selenomonas* were significantly enriched in schizophrenia patients, whereas *Eubacterium*, *Lachnospira* and *Coporcoccus* were significantly more abundant in normal controls (Fig 3D). We next performed Venn intersection analysis based on 16S rRNA gene sequencing and metagenomics screening (Fig 3E). *Clostridium* and *Megasphaera* were significantly enriched in schizophrenia patients, implicating these taxa in schizophrenia progression (Fig 3F).

## Microbial functions determined by 16S rRNA gene and metagenomic sequencing

Microbial functions were identified by PICRUSt analysis. Fig 4A shows that metabolic pathways were dominated by sphingolipid and amino acid metabolism. The other predominant KEGG pathways were associated with the cell cycle and RNA polymerase. LEfSe was performed to assess functional differences between the two groups based on metagenomic sequencing. As shown in Fig 4B, the predominant pathways were associated with fatty acid biosynthesis, phenylpropanoid biosynthesis and glycine, serine and threonine metabolism. *Bacteroides*, *Blautia*, *Clostridium*, and *Prevotella* were the primary contributors to these functions (Fig 4C).

## Discussion

Although individualized therapeutic regimens have been assessed in schizophrenia, clinical responses remain unsatisfactory. The gut microbiota has potential as a marker for infectious diseases, chronic diseases and cancer [16–18]. High-throughput sequencing enables rapid

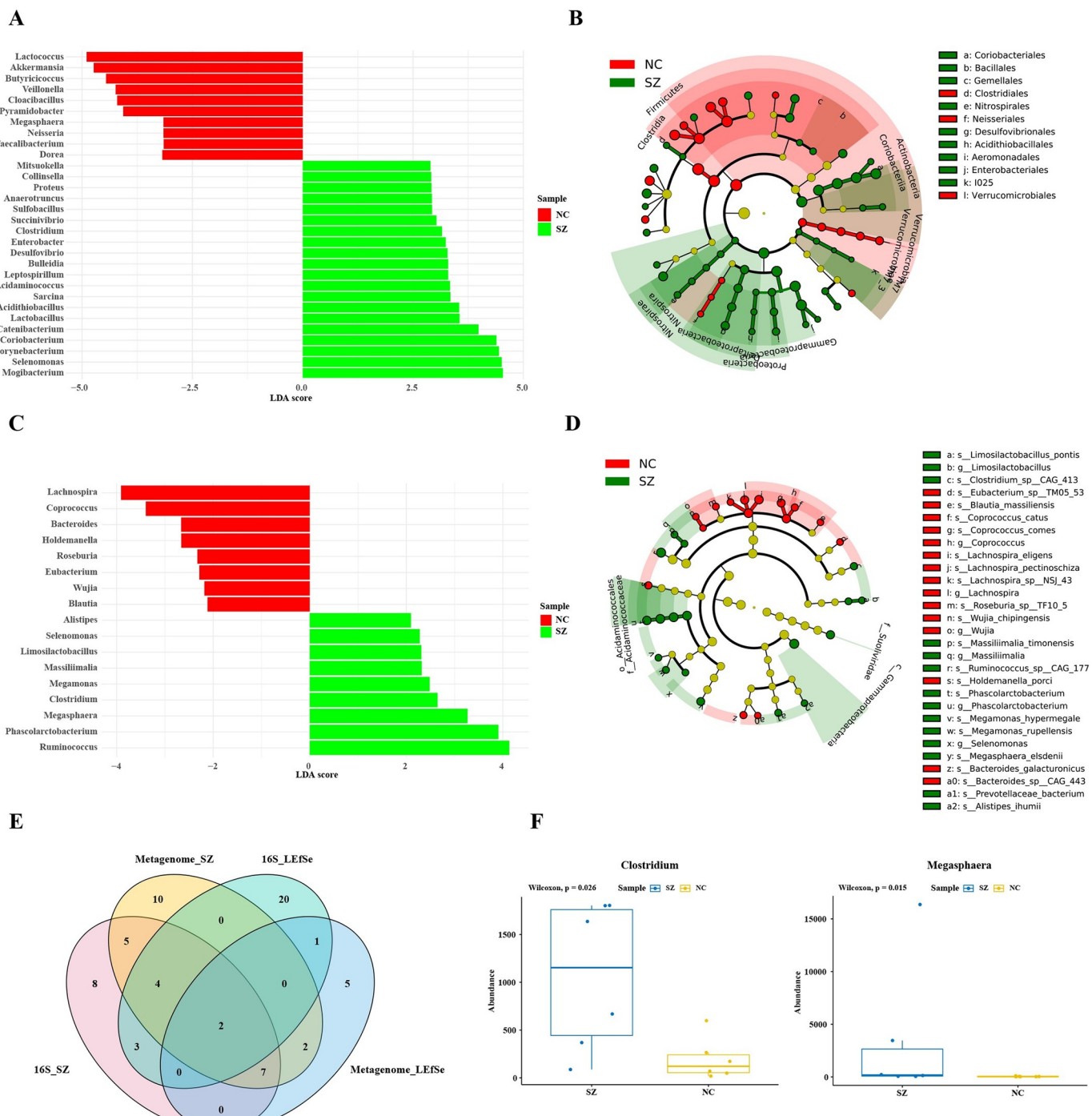

**Fig 3. Microbiota composition in groups SZ and normal control groups.** (A) LDA effect size (LEfSe) analysis was conducted to evaluate microbiota composition according to the linear discriminant analysis (LDA) score. (B) Cladogram of differential microbial composition according to LEfSe analysis. (C) LEfSe analysis on metagenomic shotgun sequencing between groups SZ and NC. (D) Cladogram of differential microbial composition by LEfSe analysis. (E) Venn analysis of predominant taxa in the gut microbiota within group SZ. (F) Relative abundances of *Clostridium* and *Megasphaera* ingroups. Note: SZ: schizophrenia; NC: normal control.

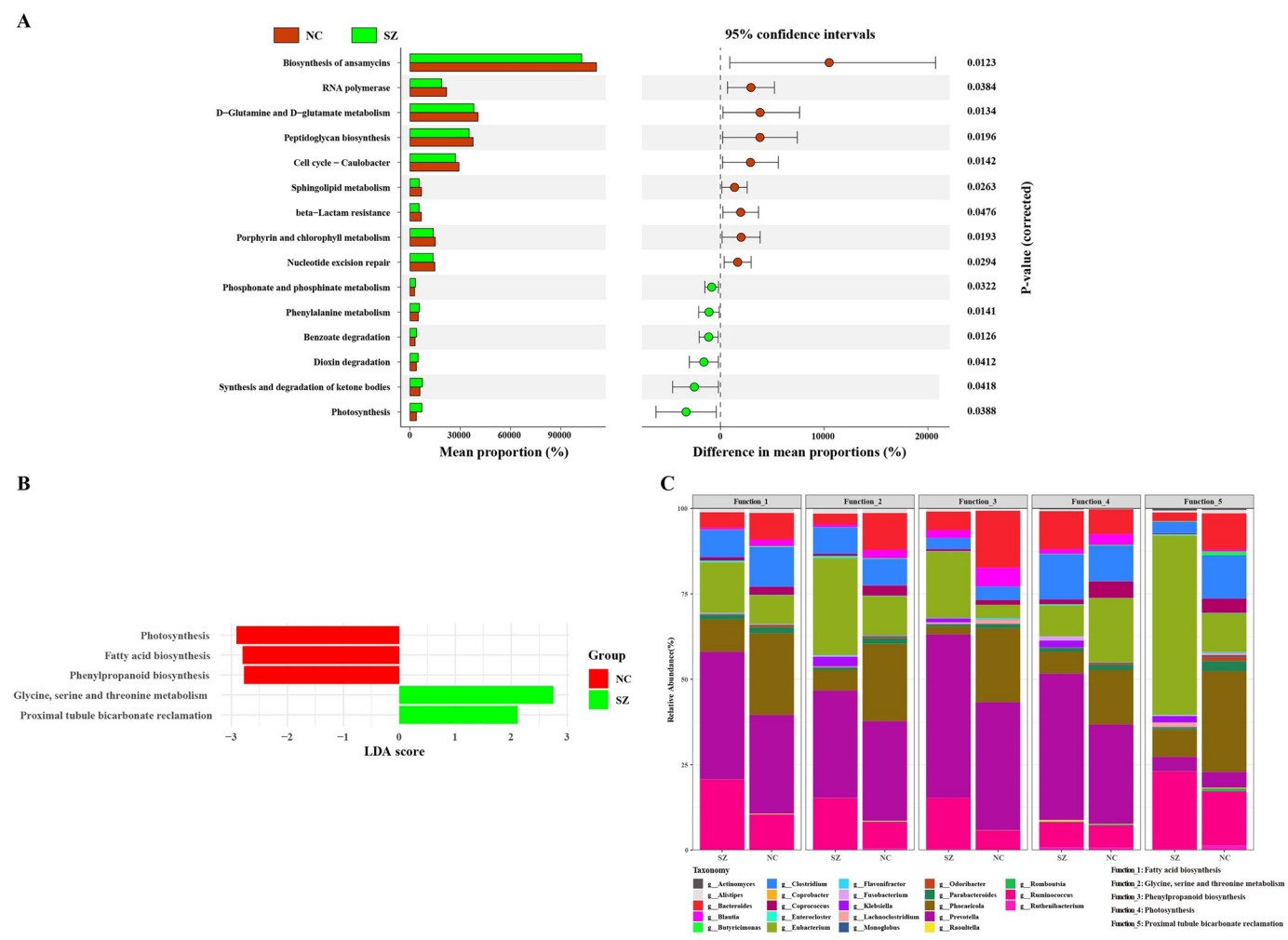

**Fig 4. Functional genes of the fecal microbiota.** (A) Differentially enriched functions between the SZ and NC groups according to phylogenetic investigation of communities by construction of unobserved states (PICRUSt) analysis. Pathways with significant difference in richness (P <0.05, computed by STAMP) between the groups are shown on the left y axis, and the P-values of enriched pathways are indicated on the right y-axis. (B) PICRUSt analysis of fecal microbiota in the SZ and NC groups based on metage metagenomics sequencing. (C) Relative contributions of genera to enriched functions. Note: SZ: schizophrenia; NC: normal control.

identification of microorganisms and differentiation of microbial community structures. However,16S rRNA gene sequencing has limitations. Taxonomic assignments rely on sequencing data from a single region of the bacterial genome, and potential primer bias may lead to disproportionate representation of some taxa. Metagenomic shotgun sequencing enables comprehensive sequencing and analysis of the entire genome, thereby providing taxonomic and functional insights. We obtained fecal samples from patients with schizophrenia, then conducted 16S rRNA gene and metagenomic shotgun sequencing to examine the relationship between the gut microbiota and schizophrenia progression. Compared with 16S rRNA gene sequencing, metagenome shotgun sequencing enables the identification of more taxa. Although alpha diversity analysis revealed minimal disparities in bacterial diversity and richness between the groups, beta diversity analysis revealed a phylogenetic distinction between patients with schizophrenia patients from normal controls. Patients with

schizophrenia showed higher abundances of *Clostridium* and *Megasphaera* according to 16S rRNA and metagenomic gene sequencing analysis.

Unidentified mechanisms potentially linked to schizophrenia were correlated with alterations in the gut microbiota, possibly induced by disruptions in the gut microenvironment [19]. Multiple bacterial taxa influence the development and advancement of schizophrenia. For example, *Clostridia* spp. modulate tyrosine expression, leading to catecholamine depletion and autism-like symptoms [20]. *Streptococcus vestibularis*, a member of the gut-brain module, synthesizes and degrades neurotransmitter such as including glutamate and GABA [21]. *Faecalibacterium* and *Roseburia* induce the production of inflammatory cytokines by disturbing host immunity [22].

In this study, the most prevalent taxa and the taxa with significantly different relative abundances in schizophrenia were consistent with prior reports, suggesting that *Clostridium* and *Megasphaera* are linked to schizophrenia group [23, 24]. In addition to *Clostridium* and *Megasphaera*, *Selenomonas*, *Ruminococcus*, and *Phascolarctobacterium* were enriched in schizophrenia group. Bacterial endotoxins, enzymes, and metabolic byproducts may alter the signaling pathways that regulate neuroendocrine immunity.

*Clostridium*, a pathogen in rectal cancer, has been linked to schizophrenia [25]. Wang Jin et al reported that *Clostridium butyicum*-induced changes in learning and memory among mice were mediated via the gut-brain axis, specifically through the alleviation of DEHP plasticizer [26]. Indeed, Yang et al. showed that an increased abundance of *Clostridium* was associated with inflammation and decreased short-chain fatty acids (SCFA) levels in an animal model [27]. *Mygasphaera*m, a Gram-negative bacterium, ferments amino acids into ammonia and branched-chain fatty acids. In the colon and cecum, *Clostridium* and *Mygasphaera*m ferment carbohydrates and amino acids; they are the primary producers of SCFAs (predominantly acetate, propionate, and butyrate). SCFAs modulate neurotransmitter transmission, regulate immune cells and inflammatory responses, function as signaling molecules, control gene expression, and provide energy. Additionally, SCFAs influence brain physiology and behavior [28]. *Mygasphaera* is associated with cognitive impairment and inflammation in patients with hepatic encephalopathy [23].

PICRUSt analysis revealed alterations in functional pathways, such as fatty acid biosynthesis, sphingolipids, glutamine and glutamate metabolism and phenylalanine metabolism. Sphingolipids are structural membrane components and important eukaryotic signaling molecules [29]. Phospholipids are precursors for synthesis of the neurotransmitter acetylcholine [30], which facilitates communication in the brain. Disruptions in the phospholipid pathway have been linked to cognitive deficits and altered behavior [31]. *Bacteroidetes* and some *Alphaproteobacteria* are prevalent in patients with schizophrenia; *Bacteroidetes* represents the sole group of gut commensal organisms capable of synthesizing sphingolipids [32]. Moreover, sphingolipids constitute a substantial proportion of the *Bacteroidetes* membrane [33]. Glutamate, an excitatory neurotransmitter, is produced by neuropod cells in the intestine and facilitates rapid signal transmission to the brain through the vagus nerve [34, 35]. Glutamate can undergo structural conversion via glutamate racemase and interactions with gut bacteria such as *Brevibacterium avium*, potentially influencing the glutamate NMDA receptor and cognitive functions in patients with Alzheimer disease [36]. Acetate, propionate, and butyrate are synthesized by commensal bacteria in the human gut and have been implicated in the effects of probiotics [37]. Phenylalanine and tyrosine are precursors for the synthesis of dopamine, a key neurotransmitter involved in the neurobiology of schizophrenia and the mechanisms of antipsychotic medications [38]. Phenylalanine and tyrosine can traverse the blood-brain barrier, and their presence in the brain modulates catecholamine synthesis [39].

In this study, we evaluated the gut microbiota of patients with schizophrenia. The 16S rRNA gene sequencing and metagenomics results showed that the abundances of *Mygasphaera* and *Clostridium* were higher in patients with schizophrenia than in normal controls. Based on these results, species diversity analysis and functional analysis both provided objective explanations concerning the development of schizophrenia. However, due to the small number of participants, the results should be verified in additional studies; investigations with larger number of participants are needed to obtain further insights concerning the effects of the gut microbiota on schizophrenia. The bacterial taxa identified in this study may be important in the etiology and progression of schizophrenia. Functional analysis revealed enrichment of sphingolipid metabolism, glutamine metabolism, and phosphonate and phosphinate metabolism in patients with schizophrenia. These findings suggest a mechanistic explanation for the sequencing results.

## Supporting information

**S1 Fig. The α-diversities of SZ and NC.** The alpha diversity between two groups, including Chao1, Simpson, Shannon, Pielou_e, Observed_species, Faith_pd, and Goods_coverage. Note: **SZ:** Schizophrenia; **NC:** Normal Controls.
(TIF)

**S2 Fig. The α-diversities of SZ and NC. (A)** Rank abundance curve. **(B)** Species accumulation curves. The β-diversities of SCZ and NC. **(C, D)**. PCoA of Unweighted and weighted Unifrac Distance at the OTU level. Note: **SZ:** Schizophrenia; **NC:** Normal Controls.
(TIF)

**S3 Fig. The distribution of Shannon diversity index of male and female.**
(TIF)

## Acknowledgments

The author thanks engineer ZhiHui Mi, TianYang Wen and LuLin Song of Inner Inner Mongolia Di An Feng Xin Medical Technology Co., LTD for providing data analysis services, Meng Han of Hulunbuir Third People's Hospital for project management service.

## Author Contributions

**Data curation:** YiMeng Wang.

**Methodology:** XiaoLong Li.

**Software:** YuTao Zhong.

**Validation:** YuTao Zhong.

**Writing – original draft:** YiMeng Wang, SiGuo Bi.

**Writing – review & editing:** SiGuo Bi, DongDong Qi.

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
