## [Decision Letter · Decision Letter 0]

13 Feb 2024

PONE-D-23-41767Altered gut microbiota in patients with schizophrenia indicate links between lipid metabolism and cognitive disorderPLOS ONE

Dear Dr. Qi,

Thank you for submitting your manuscript to PLOS ONE. After careful consideration, we feel that it has merit but does not fully meet PLOS ONE’s publication criteria as it currently stands. Therefore, we invite you to submit a revised version of the manuscript that addresses the points raised during the review process.

We look forward to receiving your revised manuscript.

Kind regards,

Christopher Staley, Ph.D.

Academic Editor

PLOS ONE

“This study was supported by grants from the Inner Mongolia Autonomous Region Science and Technology Innovation Guide project (No. 2021CXYD001) and Inner Mongolia Autonomous Region Science and Technology plan project (No. 2021GG0298).”

“This study was supported by grants from the Inner Mongolia Autonomous Region Science and Technology Innovation Guide project (No. 2021CXYD001) and Inner Mongolia Autonomous Region Science and Technology plan project (No. 2021GG0298).

The author thanks engineer ZhiHui Mi, TianYang Wen and LuLin Song of Inner Inner Mongolia Di An Feng Xin Medical Technology Co., LTD for providing data analysis services, Meng Han of Hulunbuir Third People's Hospital for project management service.

“This study was supported by grants from the Inner Mongolia Autonomous Region Science and Technology Innovation Guide project (No. 2021CXYD001) and Inner Mongolia Autonomous Region Science and Technology plan project (No. 2021GG0298).”

Reviewers' comments:

Reviewer's Responses to Questions

**Comments to the Author**

1. Is the manuscript technically sound, and do the data support the conclusions?

Reviewer #1: Partly

Reviewer #2: No

2. Has the statistical analysis been performed appropriately and rigorously? 

Reviewer #1: Yes

Reviewer #2: No

3. Have the authors made all data underlying the findings in their manuscript fully available?

Reviewer #1: Yes

Reviewer #2: No

4. Is the manuscript presented in an intelligible fashion and written in standard English?

Reviewer #1: Yes

Reviewer #2: No

5. Review Comments to the Author

Reviewer #1: In this article, Wang et al., aimed to explore the potential association between the gut microbiome and schizophrenia, utilizing 16S rRNA gene sequencing and metagenomic analysis. The study adds value to the gut-brain axis field by providing insights into the potential role of specific gut bacteria, such as Selenomonas and Clostridium, in schizophrenia. Additionally, it highlights alterations in metabolic pathways associated with the condition. This adds depth to our understanding of the complex interplay between the gut microbiome and schizophrenia pathogenesis. However, some comments are given below to consider to improve the quality of the manuscript,

1. The authors have mentioned in the abstract that “The results of the functional analysis revealed a significant correlation between the metabolism of sphingolipid, phosphonates, and phosphinates, as well as glutamine metabolism, and the manifestation and progression of schizophrenia”. However, these results need to be elaborated in the results section. Moreover, the authors must discuss the effect of phosphonates and phosphinates pathways as well. Although some potential pathways are identified, the functional mechanisms linking specific microbes to schizophrenia remain unclear.

2. The study lacks a longitudinal design, which could provide insights into the dynamic changes in the gut microbiome over time in schizophrenia patients. Moreover, very sample size especially for a metagenomics study makes statistical power and generalizability weak and unreliable.

3. The exclusion criteria for patient selection seem comprehensive but could benefit from further elaboration on how these criteria were applied and any potential biases introduced.

4. The methodology section should be more detailed, particularly regarding the data analysis pipelines for both 16S rRNA sequencing and metagenomic analysis. What are the chances of having human DNA contamination when extracted genomic DNA from fecal samples and how it was avoided?

5. The authors have limited discussion on Selenomonas. This bacterium shows significant enrichment in the schizophrenia group, therefore comparison with existing literature is needed.

6. In lines 432-433, the authors mentioned “This investigation demonstrates that patients with

schizophrenia exhibit abnormal glutamate metabolism”, however, abnormal glutamate metabolism is not mentioned in the results section. Elaborate on this point in the results and then mention it in the discussion otherwise, it is a very confusing statement.

7. As the patients were of mixed gender, please state if there is any gender-specific difference in microbiota and if is related to some specific pathway.

8. Line 449-450: Please explain how your findings provide a potential mechanism explanation for the observed sequencing data while no mechanisms have been explained.

9. Add high-quality pictures because the text in the Figures is not readable in the current format.

10. In lines 355-358, the aim of the study does not match the aim mentioned in the abstract and the objective mentioned in lines 118-119. The statement could be modified by saying that “one of the aims of this study was to investigate…..”

11. The authors have focused more on microbiota comparison. Overall, metagenomics analysis needs to be elaborated and correlated with pathways of the microbiome that respond to SZ in more detail to present clear results of your findings.

12. The article demonstrates proficiency in English language usage, with coherent sentences and technical terminology. However, there are some minor grammar issues and typographical errors that could be corrected for clarity and precision. For example, line 300: "worthing" should be "worth mentioning," and "differentially abundance" should be "differentially abundant." Line 69: “exerts a significant burden on the global population” could be “exerts a significant burden on the global health sector”. Line 210: delete one “control”; line 237: revealed or showed, delete one word; lines 428-429: Previous research??

10. All references are not in the same format. There are no details for reference numbers 6 and 7.

Reviewer #2: The authors studied gut microbiota of patients with schizophrenia using both 16s rRNA gene sequencing and metagenome, and compare it with normal control. Though interesting, the manuscript is not well organized, and many mistakes in the manuscript made it hard to follow.

Major concerns:

1. In introduction, the authors did not cite any original studies and review articles on the microbiota study of schizophrenia, which is misleading. The author should present and give credit to previous pertinent work, before highlighting what their data adds to the existing knowledge.

2. The sequencing reads number of each sample in 16s rRNA gene sequencing should be presented. And was the data normalized before analysis? Since you identified 30 unique genera in patients or control (Line 238-245), the sequencing depth must be enough to support your findings.

3. In Line 262-269, the authors compared the two groups using LEfSe. Then in Line 270-276, the authors used Wilcoxon rank-sum test to compare them again. It’s un-necessary. Same problem existed in metagenomic analysis. I suggest the authors use multiple wilcoxon ranked sign tests to compare the microbial abundances between the groups with calculation of FDR.

4. There are logical issues with metagenomic analysis. I don’t think 16s rRNA gene analysis result should be validated by metagenomic analysis. And the authors did not compare the taxa profiles obtained by this two different methods.

5. The results should be separated into several parts to make it clear and readable.

6. The title must be changed. Indeed the authors only studied gut microbiota, which can not link lipid metabolism and cognitive disorder.

Minor concerns:

Abstract

1. Line 31, ‘the of’ is wrong.

2. Change ‘16S rRNA’ to 16S rRNA gene’ throughout the manuscript.

Introduction

3. Please separate the 1st paragraph into different parts.

4. Microflora should be no longer used in the literature, please change it into microbiota.

Methods

5. Line 186, 2x250?

6. Line 193, details of the 12 participants should be given, such as xx controls and xx patients.

7. The analysis method of 16s rRNA gene sequencing is absent.

8. More details about metagenomic analysis, such as pathway prediction, should be presented.

9. The raw sequencing data should be deposited in a publicly available database.

Results

10. Line 209-210, is this a new cohort, or samples selected from the 59 volunteers?

11. Line 237, ‘revealed showed’?

12. Line 249, ’30 genera’ is not correct. It’s 198 in Figure 2A.

13. Line 251, right parenthesis is absent.

14. Line 262-265, move into method part.

15. Line 291, ‘Suoliviridae’ is not a genus name but family.

16. Line 300-302, it’s not suitable in Results, please delete it or move into Discussion.

17. Line 302-305, ‘Clostridium may exert a significant influence on the etiology and progression of schizophrenia’ is not supported by your data.

18. Line 319-324, PICRUSt is based on 16s rRNA gene prediction. But you should also present your pathway analysis based on metagenomics, which would provide strong support to this prediction.

19. Figure 1, change ‘16s rDNA gene’ into ‘16s rRNA gene’. Line 225, change ‘16S rDNA’ into ‘16s rRNA gene sequencing’. Line 226, ‘Clean reads without human sequences were blasted to NR databases.’ seems weird, please re-write this part.

20. Table 2, it’s genus, not OTU.

Discussion

1. Line 347-355, what you stated is not the truth. Please revise this part.

6. PLOS authors have the option to publish the peer review history of their article (what does this mean?). If published, this will include your full peer review and any attached files.

Reviewer #1: **Yes: **Nayla Munawar

Reviewer #2: No

---

## [Author Response · Author response to Decision Letter 0]

11 Mar 2024

Dear editor:

We feel great thanks for your professional review work on our article. As you are concerned, there are several problems that need to be addressed. According to your nice suggestions, we have made extensive corrections to our previous draft, the detailed corrections are listed below.

Reviewer #1:

Q1. The authors have mentioned in the abstract that “The results of the functional analysis revealed a significant correlation between the metabolism of sphingolipid, phosphonates, and phosphinates, as well as glutamine metabolism, and the manifestation and progression of schizophrenia”. However, these results need to be elaborated in the results section. Moreover, the authors must discuss the effect of phosphonates and phosphinates pathways as well. Although some potential pathways are identified, the functional mechanisms linking specific microbes to schizophrenia remain unclear.

A: According to your nice suggestions, we have made extensive corrections to our incorrect conclusions in the abstract section.

Q2. The study lacks a longitudinal design, which could provide insights into the dynamic changes in the gut microbiome over time in schizophrenia patients. Moreover, very sample size especially for a metagenomics study makes statistical power and generalizability weak and unreliable.

A: We agree that more studies would be useful to understand the relationships between the microbiota with the onset and progression of schizophrenia. Due to the heterogeneity of individual differences and treatment methods, no longitudinal design was conducted for enrolled cases. In the subsequent experiments, we will expand the sample cohort to provide more comprehensive data support.

Q3. The exclusion criteria for patient selection seem comprehensive but could benefit from further elaboration on how these criteria were applied and any potential biases introduced.

A: We sincerely thank the reviewer for careful reading. In order to increase the homogeneity of the study population and to reduce the risk of adverse events and bias arising from competing risks, exclusion criteria were developed to allow screening based on patients' general data. Appropriate exclusion criteria, rather than strict exclusion criteria, would enable us to obtain a sufficient number of cases for the study and more specific exclusion criteria may require more clinical evidence. We will continuously follow the latest research findings in this field and improve the inclusion and exclusion criteria for patients in subsequent studies.

Q4. The methodology section should be more detailed, particularly regarding the data analysis pipelines for both 16S rRNA sequencing and metagenomic analysis. What are the chances of having human DNA contamination when extracted genomic DNA from fecal samples and how it was avoided?

A: More detailed illustrations for both 16S rRNA sequencing and metagenomic analysis has been added in the methodology section. About phase “human DNA contamination” is a writing mistake and has been revised as follows: “deplete host DNA”.

Q5. The authors have limited discussion on Selenomonas. This bacterium shows significant enrichment in the schizophrenia group, therefore comparison with existing literature is needed.

A: The description of Selenomonas have been removed from the results and discussion section.

Q6. In lines 432-433, the authors mentioned “This investigation demonstrates that patients with schizophrenia exhibit abnormal glutamate metabolism”, however, abnormal glutamate metabolism is not mentioned in the results section. Elaborate on this point in the results and then mention it in the discussion otherwise, it is a very confusing statement.

A: The illustration of “glutamate metabolism” has been added both in the results and discussion section.

Q7. As the patients were of mixed gender, please state if there is any gender-specific difference in microbiota and if is related to some specific pathway.

A: The statistic analysis of gender-specific difference in microbiota has been added in the results section on line 266-267.

Q8. Line 449-450: Please explain how your findings provide a potential mechanism explanation for the observed sequencing data while no mechanisms have been explained.

A: The mechanisms explanation has been added in the discussion section on line 421-447.

Q9. Add high-quality pictures because the text in the Figures is not readable in the current format.

A: The figures has been revised.

Q10. In lines 355-358, the aim of the study does not match the aim mentioned in the abstract and the objective mentioned in lines 118-119. The statement could be modified by saying that “one of the aims of this study was to investigate…..”

A: According to your suggestion, we have unified this statement.

Q11. The authors have focused more on microbiota comparison. Overall, metagenomics analysis needs to be elaborated and correlated with pathways of the microbiome that respond to SZ in more detail to present clear results of your findings.

A: The illustration about the pathway has been added in the discussion section.

Q12. The article demonstrates proficiency in English language usage, with coherent sentences and technical terminology. However, there are some minor grammar issues and typographical errors that could be corrected for clarity and precision. For example, line 300: "worthing" should be "worth mentioning," and "differentially abundance" should be "differentially abundant." Line 69: “exerts a significant burden on the global population” could be “exerts a significant burden on the global health sector”. Line 210: delete one “control”; line 237: revealed or showed, delete one word; lines 428-429: Previous research??

A: We tried our best to improve the manuscript and made some changes to the manuscript.

Q13. All references are not in the same format. There are no details for reference numbers 6 and 7.

A: We have revised the reference. Reference 6 is a index of the book.

Major concerns:

Q1. In introduction, the authors did not cite any original studies and review articles on the microbiota study of schizophrenia, which is misleading. The author should present and give credit to previous pertinent work, before highlighting what their data adds to the existing knowledge.

A: We sincerely appreciate the valuable comments. We have checked the literature carefully and added reference on 12-14 into the introduction part in the revised manuscript.

Q:2. The sequencing reads number of each sample in 16s rRNA gene sequencing should be presented. And was the data normalized before analysis? Since you identified 30 unique genera in patients or control (Line 238-245), the sequencing depth must be enough to support your findings.

A: The information of sequencing reads has been added in the results section on line 242-250.

Q:3. In Line 262-269, the authors compared the two groups using LEfSe. Then in Line 270-276, the authors used Wilcoxon rank-sum test to compare them again. It’s un-necessary. Same problem existed in metagenomic analysis. I suggest the authors use multiple wilcoxon ranked sign tests to compare the microbial abundances between the groups with calculation of FDR.

A: We have rewritten this part in the results section.

Q: 4. There are logical issues with metagenomic analysis. I don’t think 16s rRNA gene analysis result should be validated by metagenomic analysis. And the authors did not compare the taxa profiles obtained by this two different methods.

A: We are sorry for our logical issues, we have made the corrections to these false claims within the whole manuscript.

Q:5. The results should be separated into several parts to make it clear and readable.

A: In the results section, we added sub-headings to make the article easier to read.

Q:6. The title must be changed. Indeed the authors only studied gut microbiota, which can not link lipid metabolism and cognitive disorder. 

A: Thank you for the title suggested. The precedent version of the title has been replaced, becoming perturbations in gut microbiota composition in schizophrenia.

Minor concerns:

Abstract

Q:1. line31, ‘the of’ is wrong 

A: We were really sorry for our careless mistakes. Thank you for your reminder.

Q:2. Change ‘16S rRNA’ to ‘16S rRNA gene’ throughout the manuscript.

A: Thanks for your careful checks. We are sorry for our carelessness. Based on your comments, we have made the corrections within the whole manuscript.

Introduction

Q:3. Please separate the 1ST paragraph into different parts.

A: Thank you for pointing this out. The 1 ST paragraph has been separated into different parts based on the statement content of each paragraph.

Q:4. Microflora should be no longer used in the literature, please change it into microbiota.

A: Thank you for pointing this out. The microflora has been corrected on microbiota.

Methods

Q:5. Line 186, 2 x250?

A: We were really sorry for our careless mistakes. Thank you for your reminder.

Q:6 Line 193, details of the 12 participants should be given, such as xx controls and xx patients

A: Thank you for pointing this out. The reviewer is correct, and we have revised it as follows “we selected a cohort of 12 participants including 6 patients with schizophrenia and 6 healthy controls”.

Q:7. The analysis method of 16s rRNA gene sequencing is absent.

A: It is really a giant mistake to the whole quality of our article. We feel sorry for our carelessness. We have corrected it and added this section in the manuscript.

Q:8. More details about metagenomic analysis, such as pathway prediction, should be presented.

A: Thank you for pointing this out. More details about metagenomic analysis have been added in the manuscript.

Q:9 The raw sequencing data should be deposited in a publicly available database. 

A: Thank you for your suggestions. All raw sequences were deposited in NCBI Sequence Read Archive under accession number PRJNA 1077638 for metagenomics sequencing and PRJNA 1077648 for 16S rRNA gene sequencing.

Results

Q:10. Line 209-210, is this a new cohort, or samples selected from the 59 volunteers?

A: More details about the participants enrolled in metagenomic sequencing section were added in the manuscript. The revised text reads as follows: We selected a cohort of 12 participants including 6 patients with schizophrenia and 6 healthy controls from 59 enrolled samples whose feces samples provided sufficient material for a metagenomic analysis.

Q: 11. Line 237, ‘revealed showed’?

A: We are really for our careless mistakes. The duplicate verbs have been removed.

Q:12. Line 249, ’30 genera’ is not correct. It’s 198 in Figure 2A.

A: As for the identification of strains, we have recognized and modified the whole process.

Q: 13. Line 251, right parenthesis is absent.

A: We are really for our careless mistakes. We have made changes to the sentences as follows: selenomonas, mogibacterium, and corynebacterium were found to be dominant with each genus comprising more than 1% of the total microbiome.

Q: 14. Line 262-265, move into method part.

A: Thank you for your suggestions. The section of “LefSe analysis” has been moved from the results section to the methods section.

Q:15. Line 291, ‘Suoliviridae’ is not a genus name but family.

A: We are really for our careless mistakes. We have made changes to the sentences as follows: family Suoliviridae.

Q:16. Line 300-302, it’s not suitable in Results, please delete it or move into Discussion.

A: We have moved this section moved into Discussion.

Q: 17. Line 302-305, ‘Clostridium may exert a significant influence on the etiology and progression of schizophrenia’ is not supported by your data.

A: We have made more appropriate changes to the sentence as follows: suggesting that Clostridium may play an important role in the progression of schizophrenia.

Q: 18. Line 319-324, PICRUSt is based on 16s rRNA gene prediction. But you should also present your pathway analysis based on metagenomics, which would provide strong support to this prediction.

A: As suggested by the reviewer, we have presented our pathway analysis based on metagenomics in the results section.

Q: 19. Figure 1, change ‘16s rDNA gene’ into ‘16s rRNA gene’. Line 225, change ‘16S rDNA’ into ‘16s rRNA gene sequencing’. Line 226, ‘Clean reads without human sequences were blasted to NR databases.’ seems weird, please re-write this part.

A: We feel sorry for our carelessness. We have re-written this part as follows: Clean reads were blasted to NR databases.

Q: 20. Table 2, it’s genus, not OTU.

A: We feel sorry for our carelessness. We have corrected the “OUT” into “genus”.

Discussion

Line 347-355, what you stated is not the truth. Please revise this part.

A: We have revised the discussion section throughout.

---

## [Decision Letter · Decision Letter 1]

14 May 2024

PONE-D-23-41767R1Perturbations in gut microbiota composition in schizophreniaPLOS ONE

Dear Dr. Qi,

Thank you for submitting your manuscript to PLOS ONE. After careful consideration, we feel that it has merit but does not fully meet PLOS ONE’s publication criteria as it currently stands. Therefore, we invite you to submit a revised version of the manuscript that addresses the points raised during the review process.

We look forward to receiving your revised manuscript.

Kind regards,

Christopher Staley, Ph.D.

Academic Editor

PLOS ONE

Reviewers' comments:

Reviewer's Responses to Questions

**Comments to the Author**

1. If the authors have adequately addressed your comments raised in a previous round of review and you feel that this manuscript is now acceptable for publication, you may indicate that here to bypass the “Comments to the Author” section, enter your conflict of interest statement in the “Confidential to Editor” section, and submit your "Accept" recommendation.

Reviewer #3: (No Response)

Reviewer #4: All comments have been addressed

2. Is the manuscript technically sound, and do the data support the conclusions?

Reviewer #3: Yes

Reviewer #4: Partly

3. Has the statistical analysis been performed appropriately and rigorously? 

Reviewer #3: Yes

Reviewer #4: No

4. Have the authors made all data underlying the findings in their manuscript fully available?

Reviewer #3: Yes

Reviewer #4: Yes

5. Is the manuscript presented in an intelligible fashion and written in standard English?

Reviewer #3: Yes

Reviewer #4: No

6. Review Comments to the Author

**Reviewer #3: **General comments:

The authors unitized 16S rRNA gene sequencing and metagenomics approach to profile the gut microbiota between schizophrenia patients and normal people. The paper has been reviewed by other reviewers and the authors have revised it, I only have several minor concerns.

Specific comments/questions:

L 41-42: based on your results, we can only conclude a relationship between gut microbiota with schizophrenia, rather than a crucial role.

L 101: collected what? This is an incomplete sentence.

L 216: why you only analysis the function of metagenomics, the diversity and composition of microbial community, even the resistome can also be obtained.

figures: the resolution of figures was low.

**Reviewer #4:** In the manuscript, the authors compared the microbial diversities, compositions, and functions between schizophrenia patients and control groups by using 16S rRNA gene sequencing and whole-genome shotgun metagenomic sequencing. They concluded that gut microbiome might exert a crucial influence on individuals with schizophrenia.

This is a carefully done study and the findings are interesting. However, some points need clarifying and certain statements require further modification. My detailed comments are given below:

1.In abstract, the authors summarized that “These findings suggest that gut microbiome may exert a crucial influence on individuals with schizophrenia” . I think more important results should be described to support this conclusion, not just “minimal variations in alpha and beta diversity” or “higher abundances of Clostridium and Megasphaera”.

2.In introduction, authors simply stated “Several studies have investigated the microbiota in schizophrenia, revealed its relationships to the onset and progression of schizophrenia”.The detailed findings of previous study are supposed to be listed.

3.In the exclusion criteria, schizophrenia patients with chronic gastrointestinal illness were not ruled out, which might affect the study results.

4.Figures in the manuscript were fuzzy. Please provide clearer figures.

5.In fig 2A, authors compared the microbial compositions between group SZ and group NC, and in abstract, authors found that “The schizophrenia patients showed higher abundances of Clostridium and Megasphaera” , however, whether the data were statistically significant were not mentioned.

6.In discussion, authors didn’t highlight the important findings of the study, analyze the reasons for observed data or compare the research findings with previous studies.

7.Limitations were not mentioned in the discussion.

8.The current manuscript needs to be polished by a native English speaker or a professional language editing service.

7. PLOS authors have the option to publish the peer review history of their article (what does this mean?). If published, this will include your full peer review and any attached files.

Reviewer #3: No

Reviewer #4: No

---

## [Author Response · Author response to Decision Letter 1]

26 May 2024

Response to Reviewers:

Reviewer 3

Q1: L 41-42: Based on your results, we can only conclude a relationship between gut microbiota with schizophrenia, rather than a crucial role.

A: We have modified this overly positive sentence. 

Q2：L 101: collected what? This is an incomplete sentence.

A: According to the reviewer’s comment, we have corrected the sentence.

Q3: L 216: why you only analysis the function of metagenomics, the diversity and composition of microbial community, even the resistome can also be obtained.

A: We are sorry for our carelessness. The section of analysis of the 16S rRNA gene sequencing has been added in the text.

figures: the resolution of figures was low.

A: Fig 1 has been redrawn, and all figures has been upload to the PACE digital diagnostic tool to meet PLOS requirements.

Reviewer 4

Q:1 In abstract, the authors summarized that “These findings suggest that gut microbiome may exert a crucial influence on individuals with schizophrenia” . I think more important results should be described to support this conclusion, not just “minimal variations in alpha and beta diversity” or “higher abundances of Clostridium and Megasphaera”.

A: We have made changes to the abstract section.

Q:2 In introduction, authors simply stated “Several studies have investigated the microbiota in schizophrenia, revealed its relationships to the onset and progression of schizophrenia”. The detailed findings of previous study are supposed to be listed.

A: Thank you for your suggestions. The detailed findings of previous study has been added in the manuscript.

Q:3 In the exclusion criteria, schizophrenia patients with chronic gastrointestinal illness were not ruled out, which might affect the study results.

A: A reference to chronic gastrointestinal illness has been added to the exclusion rule.

Q:4 Figures in the manuscript were fuzzy. Please provide clearer figures.

A: All figures have been upload to the PACE digital diagnostic tool to meet PLOS requirements.

Q:5 In fig 2A, authors compared the microbial compositions between group SZ and group NC, and in abstract, authors found that “The schizophrenia patients showed higher abundances of Clostridium and Megasphaera” , however, whether the data were statistically significant were not mentioned.

A: Statistical analysis and comparison of Clostridium and Megasphaera were shown in Fig 3F.

Q:6 In discussion, authors didn’t highlight the important findings of the study, analyze the reasons for observed data or compare the research findings with previous studies.

A: The findings has been added in the discussion section, and the comparison of the research with previous studies were illustrated on references 26-29.

Q:7 Limitations were not mentioned in the discussion.

A: Limitations of the study were added in the conclusion section of the manuscript.

Q: 8 The current manuscript needs to be polished by a native English speaker or a professional language editing service.

A: Thanks for your suggestion. We have tried our best to polish the language in the revised manuscript. The article has been sent to textcheck for polishing, and the certificate can be seen at http://textcheck.com/certificate/index/KihIW2.

---

## [Decision Letter · Decision Letter 2]

20 Jun 2024

Perturbations in gut microbiota composition in schizophrenia

PONE-D-23-41767R2

Dear Dr. Qi,

We’re pleased to inform you that your manuscript has been judged scientifically suitable for publication and will be formally accepted for publication once it meets all outstanding technical requirements.

Kind regards,

Christopher Staley, Ph.D.

Academic Editor

PLOS ONE

Additional Editor Comments (optional):

Reviewers' comments:

Reviewer's Responses to Questions

**Comments to the Author**

1. If the authors have adequately addressed your comments raised in a previous round of review and you feel that this manuscript is now acceptable for publication, you may indicate that here to bypass the “Comments to the Author” section, enter your conflict of interest statement in the “Confidential to Editor” section, and submit your "Accept" recommendation.

Reviewer #3: All comments have been addressed

Reviewer #4: (No Response)

2. Is the manuscript technically sound, and do the data support the conclusions?

Reviewer #3: Yes

Reviewer #4: Yes

3. Has the statistical analysis been performed appropriately and rigorously? 

Reviewer #3: Yes

Reviewer #4: Yes

4. Have the authors made all data underlying the findings in their manuscript fully available?

Reviewer #3: Yes

Reviewer #4: Yes

5. Is the manuscript presented in an intelligible fashion and written in standard English?

Reviewer #3: (No Response)

Reviewer #4: Yes

6. Review Comments to the Author

Reviewer #3: The authors have addressed my concerns, and I think this manuscript is ready for publication in PLoS One.

Reviewer #4: (No Response)

7. PLOS authors have the option to publish the peer review history of their article (what does this mean?). If published, this will include your full peer review and any attached files.

Reviewer #3: No

Reviewer #4: **Yes: **Yafeng Li

---

## [Editor Report · Acceptance letter]

24 Jun 2024

PONE-D-23-41767R2 

PLOS ONE

Dear Dr. Qi, 

I'm pleased to inform you that your manuscript has been deemed suitable for publication in PLOS ONE. Congratulations! Your manuscript is now being handed over to our production team.

Kind regards, 

on behalf of

Dr. Christopher Staley 

Academic Editor

PLOS ONE